# OpenReview forum: "Analysing the Generalisation and Reliability of Steering Vectors"
_NeurIPS.cc/2024/Conference — NeurIPS 2024 poster_

### Official Review · Reviewer_Z5or · 2024-07-09

**Soundness:** 3
**Presentation:** 3
**Contribution:** 3
**Rating:** 7
**Confidence:** 3

**Summary:**

This paper investigates the generalisability and stability of using steering vectors for controlling behaviour of LLMs. The authors demonstrate that steerability is highly variable across inputs and is effected by spurious concepts. The author also demonstrate the brittleness of the steering vectors to changes in prompt, resulting in poor generalisation.

**Strengths:**

1. The considered investigative study is important,and would be of great value for future research.
2. Comprehensive experimentation for validation of the claims as well as providing various intuitive explanations and possible hypothesis for observed
3. The paper is well written and easy to understand.

**Weaknesses:**

1. Some empirical results could be provided to support various hypothesis made. For example, for the case of unsteerable behaviours, whether these behaviours are linearly represented or not could be validated by some experiments rather than merely hypothesising.

**Questions:**

1. Could the authors provide some experimentation to support each of the hypothesis made regarding various behaviours similar to the one discusses in weakness 1.

**Limitations:**

Yes,

---

> ### Author Rebuttal · Authors · 2024-08-07
>
> We thank the reviewer for their sincere appreciation of our work!
>
> The reviewer’s main specific concern is about investigating whether low steerability is due to a nonlinear representation of a concept (as opposed to more mundane reasons). Generally, we think that this is a very difficult question to answer, and is largely beyond the capabilities of existing interpretability techniques. From a theoretical perspective, while we expect ‘true’ concepts to be represented linearly, the ‘true’ concepts used by a model could be very different from those used by humans. As we have no way of identifying this ground truth in language, this approach is largely intractable. From an empirical perspective, we think that the failure of steering vectors to work is sufficient evidence that the concept captured by the dataset is not linearly represented according to the standards set by other mechanistic interpretability works.
>
> One hypothesis that might explain low steerability in our specific experiments is that we optimised for the layer hyperparameter only on a specific subset of datasets, and this choice may not be optimal for the remaining datasets. To investigate this, we re-ran the layer sweep using the datasets with the lowest steerability. Please refer to Fig 4 in the rebuttal PDF. We found that these low-steerability datasets still have the same optimal layer, largely disqualifying this hypothesis. Overall, this updates us towards believing that the main reason for steering vectors not working well is that the underlying concepts are not represented linearly by the model at any layer.
>
> We are currently running additional experiments on Llama 3.1 70b, and plan to include this in the camera-ready version, but will not be able to include these results in the rebuttal version due to compute limitations. If the conclusions drawn in our paper also hold for that model, it will strengthen our argument that failure to steer is due to a nonlinear underlying representation (since we expect larger models to have better, more linear representations).
>
> We hope our response has clarified all the concerns the reviewer had, and has made them more confident in recommending the acceptance of our paper.

---

> ### Author Response · Authors · 2024-08-12
>
> Dear Reviewer,
>
> As the end of the discussion period is approaching, we want to ask if our response has addressed your concerns regarding the paper, or if you have any further questions or comments about our paper. We appreciate the time and effort put into the review process and would love the opportunity to discuss this further if our response didn't address your concerns. Thanks again for your time reviewing our paper!

---

> > ### Comment · Reviewer_Z5or · 2024-08-13
> > **Reply**
> >
> > I have read the rebuttal and my concerns are mostly addressed so I maintain my initial rating. Please make sure that the additional experiments are added to the camera-ready version.

---

### Official Review · Reviewer_h9xf · 2024-07-13

**Soundness:** 2
**Presentation:** 3
**Contribution:** 2
**Rating:** 3
**Confidence:** 3

**Summary:**

This paper primarily analyzes the generalization of the reliability of steering vectors. The authors first focus on multiple-choice questions and introduce a new metric, the steerability score, to measure the effectiveness of steering. The experimental results indicate that for many behaviors, steering is unreliable.

**Strengths:**

1. Steering vectors, as a lightweight method for controlling model behavior, have recently garnered increasing attention. This paper's focus on the generalization and reliability of steering vectors is both timely and insightful.

2. The experiments considered various behaviors and different models.

**Weaknesses:**

1. This paper focuses solely on the multiple-choice question scenario, which is just one form of data usable for calculating steering vectors. However, more practical and useful scenarios involve open-ended questions, which this paper does not analyze.

2. The authors found that the steering effects are unreliable for certain behaviors and that some biases exist. However, there is a lack of in-depth analysis regarding the factors contributing to these poor effects. For instance, the authors identified choice-bias but did not provide an explanation.

**Questions:**

N/A

**Limitations:**

Yes

---

> ### Author Rebuttal · Authors · 2024-08-07
>
> We thank the reviewer for their comments, and are glad to hear that they consider our work both timely and insightful.
>
> We agree that open-ended steering is a more practical and useful setting, and we believe our work can be generalised to this setting with minor modifications. Steerability in our paper is defined with respect to an abstract ‘propensity’ quantity. In our paper, we use the logit difference as a measure of propensity; however, other measures could be considered. In an open-ended generation setting, steerability could be considered with respect to the difference of log-probabilities (which is a generalisation of logit-difference to multiple tokens). Alternatively, it may be possible to define propensity in terms of a score given by a ‘judge’ (human or automated) that evaluates how much the targeted behaviour was elicited. A challenge with the judge score is that the resulting propensity may not be a linear function of the multiplier and may be dependent on the specific setup of the judge. Overall, we think evaluating steerability in a more general, open-ended setting (and resolving associated technical challenges) is an exciting direction for future work, and will amend our camera-ready version to include this discussion. However, this is not in the scope of this paper, which still represents significant progress towards understanding and describing the limitations of steering vectors.
>
> We also agree that it is often unclear why steering vectors do not work. Again, we believe this finding by itself has value, since previous work on steering vectors focuses on its success cases, giving an unrealistically optimistic view of steering vectors’ effectiveness. By highlighting some general failure modes, our work enables future research on investigating these failure modes in more detail, which may lead to novel insight. In addition, the novel ‘steerability bias’ that we discover is at least partially explained by spurious factors, as demonstrated in Fig 3. We think that a full explanation of ‘why’ a steering vector doesn’t work is beyond the current capabilities of the interpretability field, and that our work presents progress towards this understanding.
>
> Overall, we believe our current work has multiple valuable contributions which the reviewer has not addressed. Notably, ours is the first work to empirically evaluate steering vectors in diverse settings, highlighting important challenges for using steering vectors and challenging the prevailing belief that steering vectors ‘just work’. We reveal important and general insights about steering vectors, such as the fact that steerability seems to be a largely universal (model-agnostic) property. Finally, our work lays the foundation for analysing steerability in the open-ended setting, which we think is an exciting direction for future work. We would appreciate the reviewer commenting in more detail on these contributions.
>
> We hope our response has addressed all the concerns the reviewer had. If the reviewer has more specific concerns with our work, we would also appreciate the reviewer raising them now so that we can work to improve the paper with their valuable feedback.

---

> > ### Comment · Reviewer_h9xf · 2024-08-12
> >
> > Thank you to the authors for their detailed response. I still believe that focusing entirely on multiple-choice questions diverges too much from practical applications, making it difficult to demonstrate the role of steering vectors in more common open-ended tasks. As I mentioned in my original review, multiple-choice questions are just one method for extracting steering vectors proposed by CAA[1]. I acknowledge that evaluation on open-ended tasks is challenging, but the authors have suggested some viable methods, such as calculating the log probability of multiple tokens and using LLM-as-judge. I believe these are feasible initial attempts. Although the authors mentioned that the propensity on open-ended tasks might not have a linear relationship with the multiplier, the evaluation of multiple-choice questions in CAA is also not strictly linear. I do not think this is a significant issue.
> >
> > Additionally, while I recognize that the authors have conducted many experiments, expanding various behaviors and considering different models, the paper is positioned as the interpretability and explainability area yet does not provide insightful understandings of the failed cases, making the overall scope seem overly limited.
> >
> > Therefore, I have decided to maintain my score and suggest that the authors make revisions.
> >
> > [1] Rimsky, Nina, et al. "Steering llama 2 via contrastive activation addition." arXiv preprint arXiv:2312.06681 (2023).

---

> ### Author Response · Authors · 2024-08-12
>
> Dear Reviewer,
>
> As the end of the discussion period is approaching, we want to ask if our response has addressed your concerns regarding the paper, or if you have any further questions or comments about our paper. We appreciate the time and effort put into the review process and would love the opportunity to discuss this further if our response didn't address your concerns. Thanks again for your time reviewing our paper!

---

> ### Author Response · Authors · 2024-08-13
>
> Thank you to the reviewer for their clarifying comments.
>
> We first note that the CAA paper only proposes a methodology for extracting steering vectors from multiple-choice questions. Extracting steering vectors from open-ended prompts and completions remains an unsolved problem with no clear best approach. Among other things, it is unclear which token position to best extract the steering vector, and it has not been demonstrated that CAA will work similarly in this setting. We believe that significant methodological ablations must be done in order to first identify the best setting for open-ended steering extraction, which is not the focus of our work. Our paper's core contribution is an evaluation and analysis of the currently accepted best practice for extracting and applying steering vectors (CAA), which does not involve open-ended extraction. We would be excited to explore extracting steering vectors in an open-ended setting in future work.
>
> We acknowledge the reviewer's concern that our paper does not provide a complete explanation for the failure cases of steering vectors. Our paper would certainly be stronger if it did have such an explanation. However, we think that 'lack of steerability' is a deep and fundamental phenomenon with no simple explanation, and that finding such a complete explanation in a single work is an unreasonable bar for acceptance. Our work makes progress towards understanding the limitations of steering vectors, uncovering the phenomena of steerability bias, as well as pointing out where previous results have overestimated the usefulness of steering vectors by only focusing on the mean change rather than also examining the variance. We also provide results on the similarity of steerability across models and distribution shifts.
>
> Overall, we think that it is unrealistic to expect complete explanations for all novel phenomena in a single contribution, and that this standard neglects the value of acknowledging and critically analysing a specific problem with existing work. We note that several past interpretability papers have simply pointed out illusions or failure modes from interpretability techniques without providing a complete explanation [1-6].
>
> [1] T. Bolukbasi et al., ‘An Interpretability Illusion for BERT’, Apr. 14, 2021, arXiv: arXiv:2104.07143. doi: 10.48550/arXiv.2104.07143.
>
> [2] J. Dinu, J. Bigham, and J. Z. Kolter, ‘Challenging common interpretability assumptions in feature attribution explanations’, Dec. 04, 2020, arXiv: arXiv:2012.02748. doi: 10.48550/arXiv.2012.02748.
>
> [3] R. Geirhos, R. S. Zimmermann, B. Bilodeau, W. Brendel, and B. Kim, ‘Don’t trust your eyes: on the (un)reliability of feature visualizations’, Jun. 06, 2024, arXiv: arXiv:2306.04719. doi: 10.48550/arXiv.2306.04719.
>
> [4] J. Hoelscher-Obermaier, J. Persson, E. Kran, I. Konstas, and F. Barez, ‘Detecting Edit Failures In Large Language Models: An Improved Specificity Benchmark’, in Findings of the Association for Computational Linguistics: ACL 2023, A. Rogers, J. Boyd-Graber, and N. Okazaki, Eds., Toronto, Canada: Association for Computational Linguistics, Jul. 2023, pp. 11548–11559. doi: 10.18653/v1/2023.findings-acl.733.
>
> [5] A. Makelov, G. Lange, A. Geiger, and N. Nanda, ‘Is This the Subspace You Are Looking for? An Interpretability Illusion for Subspace Activation Patching’, presented at the The Twelfth International Conference on Learning Representations, Oct. 2023. Accessed: Aug. 13, 2024. [Online]. Available: https://openreview.net/forum?id=Ebt7JgMHv1
>
> [6] J. Miller, B. Chughtai, and W. Saunders, ‘Transformer Circuit Faithfulness Metrics are not Robust’, Jul. 11, 2024, arXiv: arXiv:2407.08734. doi: 10.48550/arXiv.2407.08734.

---

> > ### Comment · Reviewer_h9xf · 2024-08-13
> >
> > Thank you for your further response.
> >
> > 1. I am not suggesting that you explore extracting steering vectors from open-ended tasks. My point is that CAA uses multiple-choice questions as a method to extract steering vectors, the actual application scenarios are not just limited to multiple-choice questions but are focused on open-ended tasks, with many related experiments and evaluations conducted. Based on CAA's work, I am surprised that your entire paper revolves solely around multiple-choice questions.
> >
> > 2. Regarding the failure cases, I agree that fully understanding them is quite challenging, and we should not expect to completely understand this issue in a single paper. However, you have only proposed some hypotheses without any further theoretical or experimental validation.
> >
> > I understand your clarifications and will adopt an open attitude during the reviewer-AC discussion phase to ultimately assess your contribution.

---

### Official Review · Reviewer_GGDN · 2024-07-27

**Soundness:** 4
**Presentation:** 4
**Contribution:** 4
**Rating:** 8
**Confidence:** 4

**Summary:**

The authors analyze and present many problems with steering vectors.

**Strengths:**

The papers presents quite a few problems of steering vectors that deserve wide recognition.
The unreliability and propensity of steering behaviors, high variance from spurious factors, and the argumentation that steerability is a property of the dataset. To the knowledge of this reviewer, most of these problems have never been shown before.
With an area crowded with papers about slight variations/improvements (arguably pushed to satisfy investments), I find the paper refreshing in its focus on deep analysis and to promote a deep understanding of the problems in the field.

**Weaknesses:**

It would be interesting to have more experiments on some settings.
For example, expanding Fig. 7 to show more experiments that can strengthen the argumentation.
This is valid for other hypothesis/arguments too.
The abstract is rather weak, timid and non-attractive compared to the many problems raised. It is even not mentioned that SVs are mostly a "property of the dataset", among other findings.
Thus, the authors should consider revising it to better showcase the findings.

**Questions:**

Could you expand Fig 7 to verify its hypothesis further?

**Limitations:**

See above.

---

> ### Author Rebuttal · Authors · 2024-08-07
>
> We thank the reviewer for their sincere appreciation of our work! We agree that the highlighted failure modes and challenges of steering vectors deserve broad attention, and are excited for future work to analyse these failure modes in more detail.
>
> We are grateful to the reviewer for the suggestion to revise the abstract. We have adjusted the abstract to lay out our claims more clearly, and will use this new abstract in the camera-ready version of the paper. Here is the new abstract:
>
> > Steering vectors (SVs) are a new approach to efficiently adjust language model behaviour at inference time by intervening on intermediate model activations. They have shown promise in terms of improving both capabilities and model alignment. However, the reliability and generalisation properties of this approach are unknown. In this work, we rigorously investigate these properties, and show that steering vectors have substantial limitations both in- and out-of-distribution. In-distribution, steerability is highly variable across different inputs. Depending on the concept, spurious biases can substantially contribute to how effective steering is for each input, presenting a challenge for the widespread use of steering vectors. We additionally show steerability is also mostly a property of the dataset rather than the model by measuring steerability across multiple models. Out-of-distribution, while steering vectors often generalise well, for several concepts they are brittle to reasonable changes in the prompt, resulting in them failing to generalise well. Similarity in behaviour between distributions somewhat predicts generalisation performance, but there is more work needed to understand when and why steering vectors generalise correctly. Overall, our findings show that while steering can work well in the right circumstances, there remain many technical difficulties of applying steering vectors to robustly guide models' behaviour at scale.
>
> We have also extended our analysis of steerability to include Gemma 2 2b. Due to computational constraints, we were only able to run experiments on one additional model. Nonetheless, we observe that steerability remains highly correlated across models; please see Fig 6 in the rebuttal PDF. These results extend those in figure 7 to a new model and strengthen the argument that steerability is largely a universal property.
>
> We are currently running additional experiments on Llama 3.1 70b, and plan to include this in the camera-ready version, but will not be able to include these results in the rebuttal version due to compute limitations
> We hope our response has addressed all the concerns the reviewer had, and reassured them in recommending the acceptance of our paper.

---

> ### Author Response · Authors · 2024-08-12
>
> Dear Reviewer,
>
> As the end of the discussion period is approaching, we want to ask if our response has addressed your concerns regarding the paper, or if you have any further questions or comments about our paper. We appreciate the time and effort put into the review process and would love the opportunity to discuss this further if our response didn't address your concerns. Thanks again for your time reviewing our paper!

---

### Official Review · Reviewer_qr21 · 2024-07-29

**Soundness:** 3
**Presentation:** 3
**Contribution:** 3
**Rating:** 6
**Confidence:** 3

**Summary:**

The paper investigates the effectiveness of steering vectors (SVs) in guiding language model behavior by intervening the intermediate model activations at inference time. SVs have shown potential for improving model capabilities and alignment, but their reliability and generalization properties are not well understood. This study rigorously evaluates these properties, revealing substantial limitations both in-distribution and out-of-distribution.

In-distribution, steering vectors exhibit high variability across different inputs, influenced by spurious biases such as position and token biases, leading to inconsistent steerability. Out-of-distribution, while SVs often generalize well, they are brittle for several concepts and fail to adapt to reasonable changes in prompts. The generalization performance is largely a dataset property and is better when the model's behavior is similar in both source and target prompt settings.

The study extends previous analyses using a broader variety of behaviors and datasets like Model Written Evals (MWE) and TruthfulQA, employing Contrastive Activation Addition (CAA) to extract and apply steering vectors. Despite their promise, steering vectors in their current form are not reliable for aligning model behavior at scale. More work is needed to improve their reliability and generalization for practical use.

**Strengths:**

**Originality**:
The paper addresses an important gap in the literature by providing a comprehensive analysis of the generalization and reliability of steering vectors (SVs), a novel technique for guiding language model behavior at inference time. The study's focus on both in-distribution and out-of-distribution scenarios offers a deeper understanding of SVs' effectiveness and limitations, which has not been thoroughly explored in previous research.

**Quality**:
The research is methodically sound, employing a rigorous experimental design to evaluate SVs. The use of a wide range of datasets, including Model Written Evals (MWE) and TruthfulQA, ensures a thorough examination of SVs across different contexts. The paper also leverages Contrastive Activation Addition (CAA), a well-regarded method for extracting and applying SVs, ensuring the robustness of the findings. Additionally, the study provides clear metrics for assessing steerability and generalization, contributing to the reliability of the results.

**Clarity**:
The paper is well-organized and clearly written. It systematically presents the methodology, results, and implications, ensuring that readers can follow the research process and understand the findings. Detailed explanations of experimental protocols and metrics, along with visual aids like graphs and charts, enhance the clarity and comprehensibility of the study.

**Significance**:
The findings of this paper have significant implications for the field of language model alignment and behavior control. By highlighting the current limitations of SVs, the study sets the stage for future research aimed at improving these techniques. The insights gained from this research could inform the development of more reliable and generalizable methods for guiding language model behavior, ultimately advancing the state of the art in AI alignment and safety. The practical benefits of SVs, such as their potential to guide models without requiring extensive retraining or adding tokens to the context window, underscore their importance for real-world applications.

**Weaknesses:**

1. The paper focuses on multiple-choice question format with positive/negative answers only. It may be helpful to comment on how the notion of introduced steerability metric can be generalized beyond a binary setting.
2. Only two models at the 7b and 14b sizes were studied. However, as noted by the authors, these models are different sizes, architectures, and training data / algorithms yet still have results that are correlated. Investigating larger sizes (or even smaller ones / more recent ones) would strengthen the results.

**Questions:**

1. Would showing some other statistics on the steerability give any more insights? For example, showing the interquartile range / box plot (similar to Fig 3 which is only showing the mean)
2. How robust is the steerability metric introduced? For example, does it vary a lot depending on the multiplier values $\lambda$ chosen?
3. Have you considered other ways to summarize the propensity curve, instead of the mean-squares line fit? For example, we might have two propensity curves that have the same mean-squares line but one is very linear/monotonic (i.e. very steerable) while the other behaves more like the one in Figure 2. A deeper discussion justifying the proposed metric and the trade-off/caveats would be helpful.
4. Related to the above weakness, how can the introduced steerability metric can be generalized beyond a binary yes/no setting/datasets, especially in a more open-ended generation steering?

Post rebuttal: I have increased my score to 6.

**Limitations:**

The paper discussed several limitations (some which have been echoed above): going beyond MC question format, unclear how the steerability failures can be mitigated, and being limited to only 2 models for the experimental results. I think these are fair points, and would just add some more discussion as well on the proposed metric’s robustness / reliability as a metric.

---

> ### Author Rebuttal · Authors · 2024-08-07
>
> We thank the reviewer for their detailed and insightful comments! We are glad to hear that the reviewer finds our work original, clear, sound, and significant.
>
> Responding to comments raised in ‘Weaknesses’:
> 1. We define steerability with respect to an abstract ‘propensity’ quantity. In our paper, we use the logit difference as a measure of propensity; however, other measures could be considered. In an open-ended generation setting, steerability could be considered with respect to the difference of log-probabilities of a pair of desired-undesired outputs (which is a generalisation of logit-difference to multiple tokens). Alternatively, it may be possible to define propensity in terms of a score given by a ‘judge’ (human or automated) that evaluates how much the targeted behaviour was elicited. A challenge with the judge score is that the resulting propensity may not be a linear function of the multiplier and may depend on the specific setup of the judge. A different functional form (e.g. a sigmoid) could be used instead, but that would depend on the specific metric being measured. Overall, we think evaluating steerability in a more general, open-ended setting is an exciting direction for future work, and will amend our camera-ready version to include this discussion.
> 2. We agree that studying more models would strengthen the results, and have included additional results on Gemma 2 2b; please see Figure 6 in the rebuttal PDF. Due to computational constraints, we were only able to run experiments on one additional model. Nonetheless, we observe that steerability remains highly correlated across models. Overall, these results strengthen the argument that steerability is largely a universal property. We are also currently running additional experiments on Llama 3.1 70b, and plan to include this in the camera-ready version, but will not be able to include these results in the rebuttal version due to compute limitations.
>
> Responding to comments made in ‘Questions’:
> 1. We have amended our steerability plot to display the interquartile range in addition to the median; see Fig 3 in the rebuttal PDF. Overall, we notice that high-steerability datasets seem to suffer less from the steerability bias than low-steerability datasets (myopic-reward being an especially egregious example).
> 2. We have included an additional plot, ablating the steerability for different multiplier ranges; see Fig 1 in the rebuttal PDF. Overall, we find that steerability consistently increases when using a smaller multiplier; we attribute this to the fact that, closer to the 0 multiplier, the effect is likely to be more linear, whereas further out, the curve starts to look ‘S-shaped’, resulting in lower overall steerability. Generally, we think that the multiplier range is simply a hyperparameter that can be swept over; fixing a range of (-1, 1) is also reasonable. We did not optimise the multiplier range in our experiments. However, given that steerability for all datasets decreases in a consistent way, this does not affect the rank-ordering of datasets by steerability, nor would it have much effect on the correlational plots.
> 3. We agree that having more summary statistics of the propensity curve would be useful. To disambiguate between two curves having the same slope but where one is much more ‘linear’ than the other, we have also computed the mean-square error of the fit; see Fig 2 in the rebuttal PDF. We indeed find that the steering curve is typically not fully linear, as the RMSE tends to exceed what would be expected from heteroscedasticity alone. Nonetheless, we believe that calculating a slope is a reasonable summary statistic for determining overall steerability. There are downsides to the assumption of linearity, but any other functional form would require additional choices or parameters that would make the measurement less robust.
> 4. As discussed above, we believe our methodology could be generalised to open-ended steering by considering a different propensity metric; we think this is an exciting direction for future work.
>
> We plan to add all this discussion and the additional plots to the main body or the appendix of the camera-ready version of the paper.
> Overall, we’re grateful to the reviewer for their excellent suggestions in improving the paper. We hope our changes have satisfied all their concerns, and if so they will consider raising their score and recommending acceptance more strongly.

---

> ### Author Response · Authors · 2024-08-12
>
> Dear Reviewer,
>
> As the end of the discussion period is approaching, we want to ask if our response has addressed your concerns regarding the paper, or if you have any further questions or comments about our paper. We appreciate the time and effort put into the review process and would love the opportunity to discuss this further if our response didn't address your concerns. Thanks again for your time reviewing our paper!

---

> > ### Comment · Reviewer_qr21 · 2024-08-14
> >
> > Thank you for the detailed response addressing my questions and concerns, particularly on the steerability in a more general, open-ended settings, and additional results with Gemma 2 2B and several other results in the PDF. I believe that these additional results strengthen the paper and I am happy to raise my score to 6 after the rebuttal.

---

### Author Rebuttal · Authors · 2024-08-07

We thank the reviewers for their insightful comments and feedback. We are glad to see that most reviewers think our paper addresses a timely and important question, that our experiments are regarded as technically sound, and that our writing is clear and coherent.

During the rebuttal period, we have added new experimental results in support of our findings:
1. We have an updated version of Figure 3 in the main paper to also show interquartile ranges (as opposed to just the median) - see figure 1 in the rebuttal PDF. This more clearly shows the distribution of steerability scores within a dataset. These results support the claim that high-steerability datasets exhibit less steerability bias than low-steerability datasets.
2. We have extended our experimental results to include Gemma 2 2b. We found that steerability in Gemma is highly consistent with both Llama and Qwen, strengthening the argument that steering is a universal property as demonstrated in figure 7 in the main paper and figure 6 in the rebuttal PDF.
3. We are currently running additional experiments on Llama 3.1 70b, and plan to include this in the camera-ready version, but will not be able to include these results in the rebuttal version due to compute limitations
4. Addressing the concern that different datasets may be steerable at different layers, we re-ran the layer sweep on the datasets with the lowest steerability scores; we found that these datasets remained optimally steerable at the same layer, showing that the layer choice wasn’t making these datasets less steerable.
5. Addressing the question of how to choose the optimal steering multiplier, we ran an ablation on the maximum steering multiplier; we found that steerability decreases consistently as the multiplier range increases, which is expected given that the logit-diff plateaus for larger coefficients. Since the effect is uniform across all datasets, this does not affect the relative ordering of datasets by steerability, meaning that our broader conclusions still hold.

We have also identified some common concerns and suggestions raised by reviewers, which we summarise and address below.
1. On open-ended steering. Steerability in our paper is defined with respect to an abstract ‘propensity’ quantity. In our paper, we use the logit difference as a measure of propensity; however, other measures could be considered. In an open-ended generation setting, steerability could be considered with respect to the difference of log-probabilities (which is a generalisation of logit-difference to multiple tokens). Alternatively, it may be possible to define propensity in terms of a score given by a ‘judge’ (human or automated) that evaluates how much the targeted behaviour was elicited. A challenge with the judge score is that the resulting propensity may not be a linear function of the multiplier and may be dependent on the setup of the judge. Overall, we think evaluating steerability in a more general, open-ended setting is an exciting direction for future work, and will include this discussion in the camera-ready version of our work.

Overall, we are grateful to the reviewers for their detailed and insightful suggestions, and we hope our changes have satisfied all their concerns.

---

### Decision · Program_Chairs · 2024-09-25

**Decision:**

Accept (poster)

**Comment:**

I agree with the majority of the decision of the reviewers to accept the paper. I encourage the authors to consider the reviewers’ feedback to enhance the paper’s readability and incorporate the new results in the camera-ready.